# Epidemiological Impact of Metabolic Syndrome in Overweight and Obese European Children and Adolescents: A Systematic Literature Review

**DOI:** 10.3390/nu15183895

**Published:** 2023-09-07

**Authors:** Francesca Orsini, Floriana D’Ambrosio, Anna Scardigno, Roberto Ricciardi, Giovanna Elisa Calabrò

**Affiliations:** 1Laboratory of Pharmacoepidemiology and Human Nutrition, Department of Health Policy, Istituto di Ricerche Farmacologiche Mario Negri IRCCS, 20156 Milan, Italy; francesca.orsini@marionegri.it; 2Section of Hygiene, Department of Life Sciences and Public Health, Catholic University of the Sacred Heart, L.go F. Vito 1, 00168 Rome, Italy; anna.scardigno01@icatt.it (A.S.); giovannaelisa.calabro@unicatt.it (G.E.C.); 3VIHTALI (Value in Health Technology and Academy for Leadership & Innovation), Spin-Off of Università Cattolica del Sacro Cuore, 00168 Rome, Italy; robertoricciardi.mail@gmail.com

**Keywords:** metabolic syndrome, obesity, children, Europe, epidemiology

## Abstract

The prevalence of overweight and obesity is continuously increasing, both in the adult and pediatric populations, posing a substantial challenge to public health. Understanding the epidemiological burden of metabolic syndrome (MetS) among children, particularly regarding its complications and long-term effects in adulthood, is crucial for identifying effective preventive measures and enhancing the clinical care of obese children. Therefore, by searching two databases, a systematic review was conducted in order to evaluate studies that specifically addressed the epidemiological MetS impact among overweight/obese European children and adolescents. Overall, 15 studies were considered. The epidemiological data concerning the MetS impact were contingent on the diagnostic criteria used and varied across countries, resulting in a prevalence range of 1.44% to 55.8%. Spanish studies were the most numerous (34%), revealing a country prevalence rate ranging from 2.5% to 19.6%. Males (prevalence range: 1.4–55.8%) and subjects with overweight/obesity (prevalence range: 12.9–55.8%) were mainly affected. Obesity emerged as the main risk factor in the MetS development and the consequent onset of cardiovascular complications and diabetes. Knowing the MetS burden and its risk factors could improve their prevention, detection, and treatment, and guide the development of targeted public health interventions to appropriately address the health needs of younger patients.

## 1. Introduction

Today, the prevalence of overweight and obesity has become a progressively widespread issue, affecting not only adults but also the younger population.

It is estimated that approximately 39% of the global population is facing challenges related to excess weight, with a gradual rise in the prevalence of overweight conditions among children and adolescents [1].

Since 1980, the occurrence of childhood obesity has more than doubled in over 70 countries [2]. Recent data confirmed that around 38.2 million children under the age of 5 were either overweight or obese in 2019, and nearly 12.9% of children aged 5 to 9 were diagnosed as obese [1,3]. Within the European Union (EU) member states, specifically, nearly one out of every five adolescents, aged 15 years, was identified as overweight or obese on average in 2018 [4].

This trend is a significant public health concern that demands serious attention. Having excessive body weight is linked to a heightened susceptibility to cardiovascular disease (CVD), type 2 diabetes (T2DM), and metabolic syndrome (MetS), especially when body fat is localized in the abdominal area and when obesity begins in childhood [5].

Over the years, MetS, a complex disorder characterized by the coexistence of several risk factors, including dyslipidemia, central obesity, and/or insulin resistance (IR), hypertension, and dysregulated glucose homeostasis, has been extensively investigated in the adult population [6]. However, its prevalence in the pediatric population remains underestimated.

Until the present time, a global consensus on the diagnostic criteria for MetS in the pediatric population has not been reached; in most cases, the criteria applied to the adult population are adapted to children, but with different cut-off values. However, the currently available definitions of MetS (Table 1) that have been proposed, for example, by the International Diabetes Federation (IDF) [7,8], the Adult Treatment Panel III (ATP) [9], Weiss et al. [10], and the World Health Organization (WHO) [11], are not always easily applicable to pediatric patients. The lack of a common consensus on the MetS definition limits the estimation of the epidemiological burden of this condition, especially in the pediatric and adolescent populations. However, this knowledge is crucial not only to identify effective preventive strategies and enhance the clinical care of children affected by obesity but also to forecast its implication into adulthood evolution.

Hence, the primary goal of our review was to present a comprehensive synthesis of the available literature concerning the epidemiological burden of MetS among overweight and obese European children and adolescents, while also highlighting the disparities among the commonly employed definitions. Despite the absence of a universally standardized definition, our analysis aims to identify persistent gaps in the field of MetS diagnosis, as well as delineate the cluster of risk factors that predominantly contribute to its onset. This endeavor has the potential to facilitate the development of a life-course public health approach, thereby supporting the design of effective preventive strategies targeting this vulnerable young population.

## 2. Materials and Methods

A systematic literature review was carried out, and the findings were presented following the guidelines of the Preferred Reporting Items for Systematic Reviews and Meta-Analyses (PRISMA) [14]. However, the review was not registered in any review registry.

### 2.1. Search Strategy and Selection Criteria

The process of the literature search involved consulting two databases, namely PubMed and Web of Science (WoS). The following search string was used on PubMed: ((“epidemiologies” [All Fields] OR “epidemiology” [MeSH Subheading] OR “epidemiology” [All Fields] OR “epidemiology” [MeSH Terms]) AND (“metabolic syndrome” [MeSH Terms] OR (“metabolic” [All Fields] AND “syndrome” [All Fields]) OR “metabolic syndrome” [All Fields])). This search string was adapted for WoS. Filters were applied to include only English language articles, those published between 1 January 2012 and 1 April 2023, involving the target population (children: birth–18 years), and studies on humans. The search was launched on 1 April 2023.

The identified articles were organized in an Excel datasheet and subjected to a screening process based on predefined inclusion and exclusion criteria. Duplicate articles were identified and removed. The screening initially involved evaluating the titles and abstracts, followed by a review of full-text articles.

### 2.2. Study Eligibility Criteria

The review question was defined according to the PICO (Population, Intervention, Comparator, Outcomes) framework. In brief, the goal was to identify all studies investigating the epidemiological burden of MetS in children and youths affected by overweight and obesity.

Inclusion criteria were: (1) study population included children and adolescents (age range 0–18 years) with overweight and obesity, (2) original articles and systematic reviews, (3) articles written in English and (4) conducted in European countries, and (5) published between 1 January 2012 and 1 April 2023.

Exclusion criteria were: (1) only adult population (>18 years old) studied, (2) non-European contexts, (3) studies conducted in animals or in vitro, and (4) narrative reviews, commentaries, editorials, conference presentations, letters to editors, and citations without full text.

### 2.3. Data Analysis

Three researchers (F.D’A, F.O., and A.S.) independently conducted the initial screening of titles and abstracts, followed by a review of full texts. Disagreements were either resolved through discussions or by engaging a senior researcher (G.E.C.).

The information extracted from the articles that were definitively included into the literature review encompassed details such as the name of the first author, year of publication, country/setting, target population, and main findings related to MetS epidemiology in overweight and obese children and adolescents.

Upon their inclusion, the systematic reviews underwent a snowballing process that entailed utilizing the bibliographic references and citations within these reviews. This process aimed to identify additional articles that aligned with the inclusion criteria set for our review. No articles were added to our review as a result of snowballing.

The quality and applicability of the studies included were independently assessed by two researchers (F.O. and F.D’A), who used the standardized “Critical Appraisal Checklist for Analytical Cross-sectional Studies” developed by the Joanna Briggs Institute (JBI) [15]. This tool implements a checklist of eight questions on the research components and a rating scale indicating ‘Yes’, ‘No’, ‘Unknown’, and ‘Not Applicable’. Our systematic review included studies that achieved quality scores of over 50%. The researchers discussed any inconsistencies until agreement was reached, and any disagreements were solved by consulting a third reviewer (G.E.C.).

Results were summarized in narrative form and in tables and figures.

## 3. Results

### 3.1. Study Selection

The search identified 5655 records. After the removal of duplicates, 5510 articles were screened on the basis of their titles and abstracts and 476 full-text articles were selected. In adherence to the predefined inclusion and exclusion criteria, the screening process ultimately led to the inclusion of 15 articles. A comprehensive depiction of the study selection process is provided in Figure 1.

### 3.2. Study Characteristic

The 15 primary studies included in our systematic review were all cross-sectional studies [13,16,17,18,19,20,21,22,23,24,25,26,27,28,29], and no new articles were included after the snowballing process.

The studies included reported pediatric MetS data regarding European countries; specifically, 34% (*n* = 5) were conducted in Spain [19,26,27,28,29], 13% (*n* = 2) in Romania [21,22], 13% (*n* = 2) in Italy [13,24], 13% (*n* = 2) in Poland [17,18], 13% (*n* = 2) in Slovakia [23,25], 1 (7%) in Lithuania [18], and 1 (7%) in the Czech Republic [20].

The number of subjects analyzed in each study varied from 70 [18] to 2629 [23] in an age range between 5 and 19.9 years [18,24,29]. Six studies included only children who were overweight and/or obese [17,18,20,22,24], while others included subjects with variable body mass index (BMI) values.

The data regarding the epidemiological impact of MetS depended on the diagnostic criteria used, resulting in variations observed across different countries. More than half of the studies included (*n* = 11, 73.2%) based their analysis on the IDF consensus for children and adolescents [16,17,18,20,22,23,24,25,26,28,29]. One study [19] (6.7%) compared MetS prevalence obtained based on adopting two definitions of MetS: that of the National Cholesterol Education Program Program’s Adult Treatment Panel III (NCEP-ATPIII), revised by Cook/Ford [30], and that based on the IDF criteria. One study [27] (6.7%) used the MetS definition for adolescents proposed by de Ferranti et al. [12], one [21] (6.7%) adopted the criteria proposed by Weiss et al. [10], and finally, one (6.7%) introduced a new ethnicity-, age-, and gender-specific definition of MetS [13].

The main findings of this systematic review are reported below, first on analyzing the epidemiological burden of MetS among European children and youths, and then the burden of metabolic components in subjects with a diagnosis of MetS. Table 2 provides an overview of the key characteristics of each study.

### 3.3. Epidemiological Burden of MetS

In Europe, the epidemiology of MetS varies widely among countries. All the studies included in our review reported epidemiological data in terms of prevalence, except for one, which calculated the MetS incidence in children and adolescents [20]. According to the literature data analyzed, MetS mostly affects males, with prevalence ranging between 1.4% and 55.8% [16,22].

Spanish studies were the most numerous among those included (*n* = 5, 33.3%) [19,26,27,28,29]. According to the IDF criteria, these studies revealed a MetS prevalence ranging between 2.5% [26] and 19.6% [29]. In 2019, Wang et al. [26] studied 976 Spanish adolescents (45.7% males) with a mean age of 13.1 ± 1.2 years and reported a MetS prevalence of 2.5% (*n* = 24). In a sample of 976 children and adolescents aged 10–15 years enrolled by González-Jiménez et al. [28], the overall prevalence of MetS was 4.4%, with males displaying a higher prevalence (5.38%) than females (3.85%). In 2012, Civantos Modino et al. [29] analyzed a sample of 148 children and adolescents (78 boys and 70 girls) aged between 5 and 19 years. Among these subjects, 4.8% of whom were overweight and 94% were obese, the prevalence of MetS was 19.6%. Moreover, the authors found a robust correlation between hyperuricemia and metabolic disorders, documenting a MetS prevalence of 57% in subjects with uric acid levels ≥ 5.5 mg/dL and 43% in those with levels < 5.5 mg/dL.

To compare the results yielded when adopting two different definitions of MetS, Galera-Martínez et al. [19] conducted a study among 379 adolescents aged 12–16.9 years, 58.05% of whom were males. The global prevalence of MetS reported by the authors was 5.7% according to the modified NCEP-ATPIII definition, and 3.8% according to the IDF definition for adolescents, without significant differences between sexes or age groups. Finally, González et al. [27] conducted the only Spanish study that adopted the MetS definition proposed by de Ferranti et al. [12], and they reported an overall prevalence of 5.8%, which was higher in boys (10.5%) than in girls (2.7%).

In Romania, Pelin et al. [22] investigated 120 obese children aged 7 to 18, 29.2% (*n* = 35) of whom were severely obese. Although impaired metabolic components were present in 72–100% of the children enrolled, only 67 of them (55.8%) met the criteria for MetS (i.e., the presence of at least 3 of the 5 IDF criteria).

Another Romanian study, conducted by Stroescu et al. [21], analyzed the relationship between weight for gestational age and the early onset of MetS on adopting the criteria from Weiss et al. [10]. They enrolled 517 obese children, 410 (80%) of whom had been born at a weight that was appropriate for gestational age (AGA), while 107 (20%) had been small for gestational age (SGA). The authors found that the prevalence of MetS rose as age increased, both in the overall sample and when subjects were divided by age into three subgroups (pre-pubertal group, pubertal group, and adolescents). Indeed, the pre-pubertal group displayed a MetS prevalence of 4.8% in both SGA and AGA subjects; in the pubertal group, the prevalence was 7.33% in AGA subjects and 10.86% in SGA subjects, and the highest values were found among adolescents (16.8% AGA, 26.3% SGA). Thus, the main outcomes of the study were that, independently of weight status, SGA children were at higher risk of MetS, and that this increased directly with age [21].

In Poland, the prevalence of MetS was reported to range between 12.9% and 20.0% [17,18]. The more recent study, published by Jankowska et al. [17] in 2021, involved 591 obese/overweight children aged 10–12 years, and MetS was diagnosed in 12.9% of this population: 10.9% of girls and 14.6% of boys. Some years earlier, Szabelska-Zakrzewska et al. [18] had studied 70 children (58.6% females, *n* = 41) aged 5–18 years, in whom the prevalence rates of overweight and obesity were 25.7% (*n* = 18) and 74.3% (*n* = 52), respectively. The diagnostic criteria for MetS were met by 14 children (20.0%), more frequently in the 11–16 age group (*n* = 11, 78.6%), followed by the 16–18 age group (*n* = 2, 14.2%). In contrast, in the 5–10 age group, MetS was diagnosed in only one child (7.1%).

The MetS prevalence in Italy was reported in two of the studies included [13,24]. Martino et al. [13] proposed a new ethnicity-, age-, and gender-specific definition of MetS, which proved able to identify other signs and biomarkers of cardio-metabolic risk commonly associated with MetS. In 2015, the authors implemented this new definition when investigating 300 Italian subjects (51% boys) aged 6–14 years. In this target population, 13% (*n* = 38) had MetS [13].

The other Italian study, which adopted the IDF criteria, focused instead on the distribution of MetS and its components according to age-class in a total sample of 413 obese children and adolescents [24]. Overall, MetS was identified in 19.9% (*n* = 80) of subjects, with a higher prevalence (44%) in those aged less than 10 years than in the 16–20 age group (18.9%) and the 10–16 age group (11.4%).

In Slovakia, Jurkovičová et al. [23] analyzed a sample of 2629 adolescents (54.2% females) aged between 14 and 18 years, in which a significantly higher prevalence of overweight and obesity was recorded in males (30.7% vs. 22.9%). In the whole sample, MetS was diagnosed in 1.44% (*n* = 38) of adolescents, with a higher prevalence among boys (2.7%, *n* = 32) than among girls (0.4%, *n* = 6). Focusing on the overweight/obesity group (*n* = 696) enrolled in the study, the authors estimated a MetS prevalence of 5.5%, which was higher in boys than in girls (8.6% vs. 1.8%). In another study conducted in Slovakia, during the 10-year period from 2003 to 2012, the observed overall prevalence of MetS was 2.75% among 1294 children aged under 18 years (2.8% of females; 2.7% of males) [25].

The last two studies included in this systematic review involved obese pediatric populations from Lithuania and the Czech Republic [16,20].

In Lithuania, Smetanina et al. [16] studied 344 obese/overweight children and adolescents aged 10–17 years in all stages of puberty and found MetS in about one-fifth of the sample (*n* = 73, 21.3%). The gender distribution of MetS displayed a higher prevalence in obese/overweight boys than in obese/overweight girls. Moreover, the MetS prevalence was significantly higher in obese girls than in overweight girls (overweight group: 8.7% girls vs. 20% boys; obese group: 40.8% girls vs. 52.5% boys).

Finally, the study by Pastucha et al. [20] was the only one that reported MetS data in terms of incidence. On analyzing 274 obese Czech children aged 10–17 years, the authors estimated an incidence of 37% (*n* = 102) in the whole sample.

### 3.4. Prevalence of Different Components of MetS

Six studies included in our review (6/15, 40%) investigated the distribution of metabolic components of MetS in children and adolescents [13,16,17,18,22,28].

In a group of 14 children with MetS, Szabelska-Zakrzewska et al. [18] frequently observed a decrease in HDL-C levels (92.8%, *n* = 13) and an increase in TG concentrations (71.4%, *n* = 10), in addition to abdominal obesity in 100% of cases. Furthermore, arterial hypertension was found in 71.4% (*n* = 10) of MetS children, while an increase in serum glucose was detected in only one child (7.1%). Similarly, Jankowska et al. [17] registered reduced levels of HDL-C in all obese Polish children suffering from MetS (*n* = 76, 100%) and increased TG levels in more than half (55.3%). Elevated blood pressure was also detected in 73% of children diagnosed with MetS. Pelin et al. [22] also emphasized the relevance of dyslipidemia and hypertension as risk factors for MetS. In 67 Romanian MetS children aged 7–18 years, they found low HDL-C levels, high TG, and high BP in 100%, 48%, and 50% of the sample, respectively.

In Spain, González-Jiménez et al. [28] observed elevated mean values of the variables of abdominal obesity, blood pressure, and basal glucose in both sexes in a group of 43 adolescents diagnosed with MetS. Regarding increased TG (*n* = 8, 34.8%) and low levels of HDL-C (*n* = 2, 8.7%), a higher prevalence was found in males.

On analyzing the components of MetS in Lithuania, Smetanina et al. [16] found that one in six obese or overweight children or adolescents aged 10 to 17 years had lipid metabolism abnormalities. Specifically, of 73 overweight/obese children or adolescents with a MetS diagnosis, 45.7% (*n* = 33) had increased TG levels, and 65.7% (*n* = 48) had decreased HDL-C. Furthermore, elevated systolic or diastolic blood pressure was observed in 60.0% and 41.4% of these subjects, respectively.

Finally, in the Italian study by Martino et al. [13], 38 children (13%) with metabolic disorders presented: elevated systolic blood pressure (82.09%), low diastolic blood pressure (73.13%), hyperglycemia (56.72%), high TG levels (55.22%), AO (37.31%), and low HDL-C (23.88%).

### 3.5. Quality Assessment

In order to objectively evaluate the 15 studies, we used the JBI critical appraisal checklist, and the results are presented in Table 3. None of the studies were excluded based on the results of the quality assessment. All studies obtained a “yes” answer to at least five of the eight items. All the studies included provided adequate information about their participants, the study setting, the details of the exposure measured, and the criteria used to measure the condition and outcomes. In all the studies, the assessment of confounding factors and strategies for dealing with these was “not applicable”. Finally, only 53.3% (*n* = 8) of the studies clarified the statistical analyses employed.

## 4. Discussion

This systematic review provided a comprehensive overview of the epidemiological burden of MetS in overweight and obese European children and adolescents, while also comparing values obtained through different adopted definitions.

According to the data analyzed, MetS epidemiology varied widely among European countries, and was reported mainly in terms of disease prevalence. Indeed, only one study reported the MetS incidence in European children and adolescents [17]. From the studies included in our review, it emerged that the MetS prevalence varied between 1.44% and 55.8% in the European pediatric population [16,22], mostly affecting males [16,17,19,20,23,27,28]. However, it is crucial to emphasize that the studies encompassed in our review employed varying definitions of MetS. Indeed, to date, there are no unequivocal international criteria for the MetS diagnosis in the pediatric population. Thus, as the different MetS definitions applied in children have different sets of cut-off values, the estimated prevalence of MetS differs [5]. The definition proposed by the IDF [8] is the most effective and easy to use, as well as being the one most widely used in clinical practice [5]. On applying the IDF criteria, the Spanish studies yielded a MetS prevalence ranging from 2.5% in adolescents with a mean age of 13.1 ± 1.2 years [26] to 19.6% in children and adolescents aged between 5 and 19 years [29]. In contrast, in the study by Galera-Martínez et al. [19], the MetS prevalence in Spanish adolescents was 5.7% according to the modified NCEP-ATPIII definition [9], and 3.8% according to the IDF definition. On adopting the MetS definition proposed by de Ferranti et al. [12], González et al. [27] reported an overall prevalence of 5.8%.

In Poland, the MetS prevalence, when calculated according to the IDF criteria, ranged from 12.9% in children aged 10–12 years to 20.0% in children/adolescents aged 5–18 years [17,18].

In Italy, the MetS prevalence ranged from 13% in children/adolescents aged 6–14 years, according to the Martino criteria [13], to 19.9% in children/adolescents aged 5–20 years, according to the IDF criteria [24].

In Slovakia, the MetS prevalence according to the IDF criteria ranged from 2.75% in children under 18 years of age [22] to 5.5% in adolescents (14–18 years) with overweight and obesity [23].

Of the studies included in our review, only one, conducted in the Czech Republic, reported incidence data [20]. In a group of 274 obese Czech children/adolescents aged 10–17 years, a MetS incidence of 37% was estimated (according to the IDF criteria) [20].

Although the data on the prevalence of MetS in the overweight and obese pediatric populations varied among the different countries considered, the different age groups, and, above all, the diagnostic criteria used, the most worrying fact is that this syndrome is frequent in European obese children/adolescents. Obesity, and in particular severe obesity, constitutes a significant public health concern, impacting a substantial number of children across Europe [31]. A recent study encompassing 21 European countries [31] highlighted significant variations in the prevalence of severe obesity across these countries, particularly noting higher rates in Southern Europe. This elevated prevalence was commonly observed among boys as compared to girls, and in several countries, one in every four obese children were classified as severely obese.

Countries within the Mediterranean basin have reported the highest rates of overweight and obesity, ranging from 24% to 37% [32]. In contrast, Atlantic and Central European countries exhibit considerably lower prevalence rates, typically ranging from 14% to 21% [32]. Specifically, Italy (32.4%; 95% CI: 23.8–42.4), Greece (29.6%; 95% CI: 14.5–45.0), and Portugal (26.4%; 95% CI: 23.8–29.2) have documented the highest prevalence estimates. Conversely, Estonia (8.3%; 95% CI: 6.6–10.5), France (11.0%; 95% CI: 7.7–15.4), and The Netherlands (13.4%; 95% CI: 12.5–14.3) report the lowest prevalence range of overweight and obesity [32,33].

Extrapolating the prevalence estimates based on the WHO definition [34] to encompass the entire population of children aged 6–9 years in each European country, it is projected that approximately 398,000 children would meet the criteria for severe obesity within these 21 countries.

However, in recent decades, the growing prevalence of obesity among children and adolescents, as well as adults, has evolved into a substantial health challenge worldwide.

In 2020, it was estimated that within the WHO European Region, approximately 4.4 million children under the age of 5 were impacted by overweight, including obesity, with significant disparities among different countries [35].

The prevalence of pediatric obesity, in particular, rose dramatically from 4% in 1975 to 18% in 2016, with an increase from 5 million to 50 million girls and from 6 million to 74 million boys [36,37]. An age-based analysis in 2016 showed that 40 million children under the age of 5 years and more than 330 million children and adolescents aged 5–19 years were overweight or obese [38,39]. In the same year, an estimated 216 million were classified as overweight [36]. Aside from high-income nations [36], the obesity epidemic has been growing most rapidly also in low- and middle-income countries, notably in regions such as Northern and Southern Africa, the Middle East, and the Pacific Islands [40].

In Africa, for instance, the count of overweight children under the age of 5 has surged by nearly 24% since 2000, and almost half of the children under 5 who were overweight or obese in 2019 lived in Asia [1]. In the USA, obesity rates have shown a consistent increase since 1999–2000, reaching a prevalence of 18.5% among children and adolescents in 2015–2016. [41,42].

Indeed, despite the heterogeneous patterns observed at national levels, the rising obesity prevalence represents one of the most pressing public health concerns of our era, affecting nations worldwide [40].

According to several studies, 80% of adolescents aged 10 to 14 years, 25% of children younger than the age of 5 years, and 50% of children aged 6–9 years with obesity are at risk of developing obesity that persists into adulthood [42,43]. Furthermore, during a 20-year follow-up, adolescents with a BMI > 25 kg/m^2^ reported an almost two-fold increase in the risk of death in adulthood compared to individuals with a BMI < 25 kg/m^2^ [42,44]. Thus, the excess of body fat during childhood increases the risk of obesity in adulthood five-fold and is associated with serious cardio-metabolic complications, including dyslipidemia, hyperglycemia, and hypertension [45]. This set of risk factors defines MetS and increases the risk of future CVD and T2DM [5].

Similarly, within the studies we examined, MetS was most commonly observed among obese adolescents, displaying a prevalence of 32.1%, in contrast to a lower prevalence of 7.1% in overweight adolescents [46]. In 2020, Calcaterra et al. [47] detected MetS only in obese subjects (14.27%), confirming that MetS is a main consequence of obesity, particularly in the post-pubertal period. Moreover, our results revealed an extremely worrying trend, where health risks linked to MetS manifest early in childhood and adolescence, exhibiting a substantial correlation with an elevated probability of encountering chronic diseases in adulthood [48,49].

Only six of the included studies in this systematic review concerned the prevalence of distinct components of MetS. Four out of the six reported that dyslipidemia and hypertension were the most frequent anomalies in overweight and obese children from Poland [17,18], Romania [22], and Lithuania [16]. The other two studies, conducted in Spain and Italy, also reported impaired glucose metabolism [13,28].

All these components could hold prognostic significance for the development of MetS, and their early detection might facilitate preventive measures, reducing its prevalence and related consequences [1]. However, the absence of universally recognized and standardized criteria for both MetS and its components complicates the comprehensive management of MetS’s impact during childhood [6]. Thus, the establishment of clear definitions and diagnostic thresholds for each MetS component, tailored to the pediatric population, is a critical step for accurate diagnosis, timely interventions, and effective preventive strategies.

Moreover, the development of these criteria should take into account additional variables such as age, gender, pubertal stage, and ethnicity [50], encompassing distinct biological and developmental attributes as well as environmental influences affecting children and adolescents. For instance, several anthropometric measurements have been suggested and employed to predict the presence of obesity among the pediatric population, including weight-, age-, and gender-specific centiles, BMI, BMI standard deviation score (BMI-SDS), waist circumference (WC), and waist-to-hip ratio (WHR) [51,52]. However, it is important to note that an individual’s ethnicity can significantly influence the definition of cut-off points for these indicators.

Similarly, established cut-off values for lipid profiles, blood pressure, and fasting glucose levels in adults may not always be directly applicable to adolescents without considering factors such as pubertal stage, gender, ethnicity, and clinically relevant conditions [50]. Therefore, to enhance the accuracy and effectiveness of MetS assessment and management among children and adolescents, it would be advisable to consider the influence of all the individual characteristics for the development of specific marker cut-off points. Previous research has also highlighted racial and ethnic disparities in individual MetS components, underlining the role of genetic factors and the potential variability of MetS presentation among different subgroups [53,54,55]. Although further evidence is needed, a thorough exploration of these factors will expedite the timely identification of MetS cases, allowing for swift interventions and personalized preventive strategies.

To date, the most effective strategy to mitigate the future prevalence of MetS is to proactively prevent obesity among children and adolescents. Additionally, early detection of MetS could empower pediatricians to promptly recognize and manage children who are at a heightened risk of cardio-metabolic complications. This proactive approach could consequently contribute to the reduction of CVD and T2DM incidence in adulthood [5].

Due to its far-reaching consequences on health, social welfare, and economic systems, addressing MetS necessitates a comprehensive array of strategies. These encompass early prevention efforts targeting overweight and obesity, as well as targeted interventions for those requiring treatment. In this regard, it is essential to promote a healthy and active lifestyle starting early in life. This involves not only sustaining adequate levels of physical activity among younger children [56], but also enhancing nutritional education for both children and their parents. This emphasis on nutritional education is particularly pertinent within the school setting [57].

Implementation of the Mediterranean diet, which involves consuming plenty of vegetables and olive oil, is also important. For example, in a 16-week study of adherence to the Mediterranean diet among children and adolescents, the MetS prevalence decreased (from 16% to 5%) among those on the diet, while it did not change, or even worsened, in the control group [58].

The most impactful strategies are likely those that integrate both reducing calorie consumption and boosting energy expenditure. Additionally, there should be heightened endeavors to combat obesity and MetS through a public policy lens, which involves creating secure spaces for physical activity and advocating for nutritious options within the school setting [59].

Our review provides evidence on the epidemiological burden of MetS in European children and youths and underlines the main risk factors, such as overweight and obesity, associated with the syndrome in this target population. It also indicates that clinicians should identify at-risk patients and provide guidance on the prevention and management of this disease. Comprehensive information on the prevalence of MetS can, therefore, support value-based decision-making aimed at identifying adequate strategies for the prevention of this disorder, which, if not managed, can lead to serious complications in adulthood.

### Strengths and Limitations

Through a comprehensive search strategy, this review has provided significant insights into the prevalence of MetS among children and adolescents in Europe.

Knowing the epidemiological impact of MetS in this young population, especially concerning the complications associated with this condition and its evolution in adulthood, is essential for identifying adequate preventive strategies and improving the clinical management of children affected by obesity.

Furthermore, the identification of primary risk factors associated with the syndrome, particularly the strong connections to overweight and obesity, not only enhances the comprehension of this condition but also constitutes a crucial step in formulating tailored approaches and initiatives aimed at mitigating its consequences. Notably, our results could support informed decision-making processes in public health planning and intervention efforts, thus ensuring the protection of the health and well-being of the younger generation. At the same time, by highlighting the gaps in existing research, our study underscores the need for further investigation in specific areas, such as the identification of optimal diagnostic criteria for children and adolescents and the exploration of MetS’s implications across different demographic groups.

However, several limitations should be considered when interpreting our findings. Only articles published in English were included, which might have prevented us from identifying all the available evidence on the topic. While we rigorously adhered to the PRISMA statement during the screening process, it is important to acknowledge that a potential for selection bias remains, albeit not entirely eliminated [60]. Moreover, although we performed a comprehensive search strategy, the studies’ heterogeneity, in terms of study-specific characteristics, including study design, study populations, and the different criteria used for the MetS diagnosis, prevented us from issuing precise estimates of the epidemiological burden of childhood MetS in Europe. Additionally, all the studies included in our review reported epidemiological data in terms of prevalence, except for one, which calculated the incidence of MetS in children and adolescents, and this limits estimation of the complete epidemiological burden of MetS in this target population.

Nevertheless, our overview pinpointed several meaningful aspects, which can help in the appraisal of preventive interventions targeted to overweight and obese children and adolescents at risk of MetS.

## 5. Conclusions

Obesity is a major risk factor for MetS development and the consequent onset of cardiovascular complications and diabetes. Although multidisciplinary lifestyle intervention has proved most efficacious [59], investment in scientific research should also be increased. Such research should be aimed at identifying new therapeutic targets for MetS, even in children, and at producing further scientific evidence on the new treatments that have been proposed in recent years, such as the Polysaccharide-Based Natural Substance Complex, which is capable of acting simultaneously on two or more components of MetS [61].

Another challenge is how to intervene at the public health level in order to reduce the high prevalence of obesity in the general population. There is widespread consensus within the scientific community that combating the obesity epidemic requires a shift away from individual behavior-focused prevention and towards strategies that are rooted in the environment or community. This approach would offer the greatest possibility of reducing MetS, making the role of the pediatrician crucial to both the diagnosis and prevention/treatment of obesity and all its associated risks and/or complications.

## Figures and Tables

**Figure 1 nutrients-15-03895-f001:**
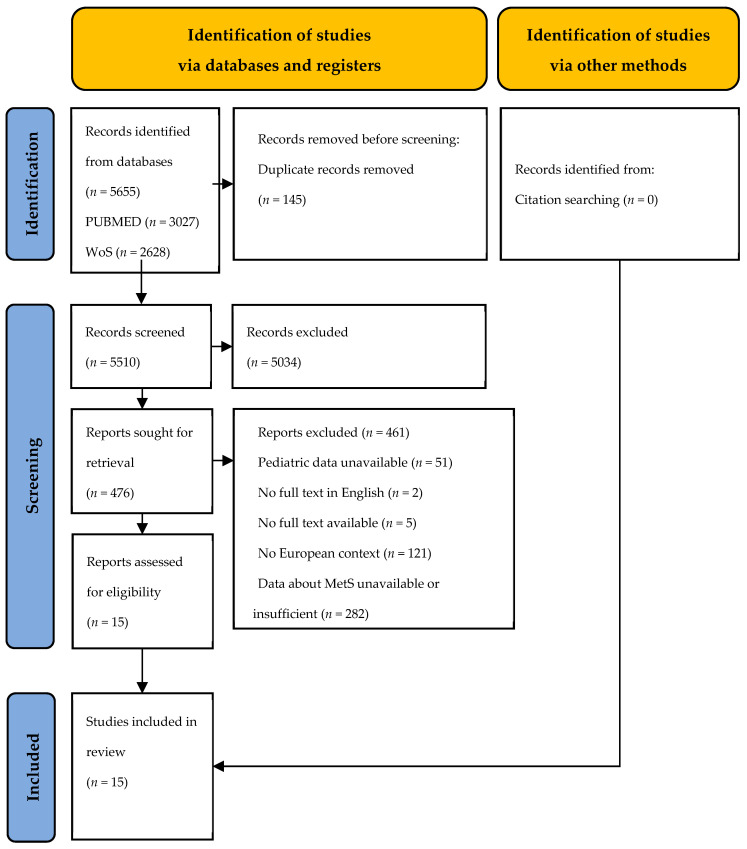
PRISMA flow chart of the inclusion process.

**Table 1 nutrients-15-03895-t001:** Definitions currently used for the MetS diagnosis in children.

	IDF [8]			ATP [9]	WHO [11]	Weiss et al. [10]	de Ferranti et al. [12]	Martino et al. [13]
Age group (years)	<10	10–16	>16	-	-	-		
Criteria	AO + two or more of the four criteria			Any three of the five criteria	GI * + two or more of the other components	Any three of the five criteria		
Abdominal obesity (AO)	≥90th percentile (WC)		WC: ≥94 cm for European men, ≥80 cm for European women, with ethnicity-specific values for other groups	≥90th percentile (WC)	WHR > 0.9 in men, >0.85 in women, and/or BMI > 30 kg/m^2^	BMI z-score ≥ 2	>75th percentile (WC)	>90th percentile (WC)
Triglycerides (TG)	**	≥1.7 mmol/L (≥150 mg/dL)				≥95th percentile	≥1.1 mmol/L (≥100 mg/dL)	>90th percentile
HDL-cholesterol (HDL-c)	**	<1.03 mmol/L (<40 mg/dL)	<1.03 mmol/L (<40 mg/dL) in men <1.29 mmol/L (<50 mg/dL) in women or in treatment for lipid abnormalities		<0.91 mmol/L (<35 mg/dL) in men <1.01 mmol/L (<39 mg/dL) in women	≤5th percentile	<1.3 mmol/L (boys aged 15–19 years, <1.17 mmol/L)	<10th percentile
Blood pressure (BP)	**	Systolic ≥ 130/diastolic ≥ 85 mmHg or in treatment for hypertension		≥90th percentile (age-, sex-, and ethnicity-specific)	Systolic ≥ 140 Diastolic ≥ 90 mmHg	≥95th percentile	>90th percentile	
Fasting glucose levels	**		≥5.6 mmol/L (100 mg/dL) or known diabetes mellitus		≥6.1 mmol/L (110 mg/dL), subsequently lowered to ≥5.6 mmol/L (100 mg/dL) *	GI (ADA criteria)	≥6.1 mmol/L (110 mg/dL)	>90th percentile
Microalbuminuria	-				Urinary albumin excretion rate ≥ 20 μg/min or albumin/creatinine ratio ≥ 30 mg/g	-	-	-

Abdominal obesity (AO), American Diabetes Association (ADA), body mass index (BMI), glucose intolerance (GI), high-density lipoprotein (HDL), International Diabetes Federation (IDF), National Cholesterol Education Program Adult Treatment Panel III (ATP), World Health Organization (WHO), waist circumference (WC), waist-to-hip ratio (WHR), * or impaired glucose regulation or diabetes mellitus and/or insulin resistance. ** The diagnosis of MetS requires the presence of disturbances in these parameters within the family history. Source: Modified from Grabia M et al., 2021 [7].

**Table 2 nutrients-15-03895-t002:** Data extracted from the studies encompassed in our systematic review.

First Author, Year	Country	Sample Size Total (F/M)	Age (Year)	Overweight (Ow)	Obesity (Ob)	Diagnosis of MetS		Components of MetS				
						Total	F/M	Ab. Obesity	Low HDL	High TG	High BP SBP/DBP	Glycemia
		N.	Min–Max	N (%)	N (%)	N (%)	N (%)	N (%)	N (%)	N (%)	N (%)	N (%)
Smetanina, N., 2021 [16]	Lithuania	344 (193/151) OwOb	10–17	60 (10.7%) F: 49 (83.1%) M: 11 (16.9%)	284 (82.5%) F: 144 (50.7%) M: 140 (49.3%)	73 (21.3%) (IDF)	Ow: F: 8.7% M: 20% Ob: F: 40.8% M: 52.5%	n/d	48 (65.7%)	33 (45.7%)	54 (74.3%)/ 60 (41.4%)	n/d
Jurkovičová, J., 2021 [23]	Slovakia	2629 (1424/1205)	14–18	426 (16.9%) F: 214 (15.0%) M: 212 (17.6%)	270 (10.27%) F: 112 (7.9%) M: 158 (13.1%)	38 (1.44%) (IDF)	F: 6 (0.4%) M: 32 (2.7%)	n/d	n/d	n/d	n/d	n/d
Jankowska, A., 2021 [17]	Poland	591 (275/316) Obese	10–12	401 (67.9%) F: 190 (69.1%) M: 211 (66.8%)	190 (32.1%) F: 85 (30.9%) M: 105 (33.2%)	76 (12.9%) (IDF)	F: 30 (10.9%) M: 46 (14.6%)	76 (100%)	76 (100%)	42 (55.3%)	56 (73%)	26 (34.2%)
Leone, A., 2020 [24]	Italy	413 (244/169) Obese	7–19.9 Age range groups: 7–9.9 (n. 84) 10–15.9 (n. 229) 16–19.9 (n. 90)	n/d	403 (100%)	80 (19.85%) Age range groups: 7–9.9: 37 (44%) 10–15.9: 26 (11%) 16–19.9: 17 (18.9%) (IDF)	n/d	n/d	n/d	n/d	n/d	n/d
Wang, J., 2019 [26]	Spain	976 (519/457)	10–15	n/d	n/d	24 (2.5%) (IDF)	n/d	n/d	n/d	n/d	n/d	n/d
Szabelska-Zakrzewska, K., 2018 [18]	Poland	70 (41/29) OwOb	5–18	18 (25.7%)	52 (74.3%)	14 (20%) (IDF) Age range groups: 5–10: 1 (7.1%) 11–16: 11 (78.6%) 16–18: 2 (14.2%)	n/d	Ow: 3 (21.4%) Ob: 11 (78.6%)	Ow: 3 (21.4%) Ob: 10 (71.4%)	Ow: 3 (21.4%) Ob: 7 (50.0%)	Ow: 1 (7.1%) Ob: 9 (64.3%)	Ow: - - Ob: 1 (7.1%)
Ostrihoňová, T., 2017 [25]	Slovakia	1294 (−/−)	10-17	n/d	n/d	36 (2.78%) (IDF)	F: 21 (2.8%) M: 15 (2.7%)	n/d	n/d	n/d	n/d	n/d
Galera-Martínez, R., 2015 [19]	Spain	379 (159/220)	12–16.9	F: 18.3% M:23.6%	F: 9 (5.7%) M: 21 (9.7%)	5.7% (NCEP-ATPIII) 3.8% (IDF)	F: 3.2% M: 7.6% (NCEP-ATPIII) F: 1.9% M:5.2% (IDF)	n/d	n/d	n/d	n/d	n/d
González-Jiménez, E., 2015 [28]	Spain	976 (519/457)	10–15	n/d	F: 120 (23.12%) M: 122 (26.69%)	43 (4.4%) (IDF)	F:20 (3.85%) M:23 (5.38%)	F: 20 (100%) M: 23 (100%)	F: - (0%) M: 2 (8.7%)	F: - (0%) M: 8 (34.8%)	SBP: F: 17 (85%) M: 21 (91,3%) DBP: F: 5 (1%) M: - (0%)	F: 11 (55%) M: 9 (39.1%)
Martino, F., 2015 [13]	Italy	300 (147/153)	6–14	n/d	n/d	38 (13%) (Martino 2015 [13])	n/d	37.31%	23.88%	55.22%	SBP: 82.09% DBP: 73.13%	56.72%
Pastucha, D., 2014 [20]	Czech Republic	274 (128/146) Obese	10–17	n/d	274 (100%)	102 (37%) (IDF) (incidence)	F: 31% M: 43% (incidence)	25.27% (incidence)	0.25% (incidence)	0.86% (incidence)	SBP: 14.69% DBP: 11.37% (incidence)	0.52% (incidence)
Stroescu, R., 2014 [21]	Romania	517 (235/282) Obese SGA: 107 (20%) AGA: 410 (80%)	Pre-pubertal (5–10) SGA: 42 AGA: 165 Pubertal (11–14) SGA: 46 AGA: 150 Adolescents (15–18) SGA: 19 AGA: 95	n/d	517 (100%)	4.8–26.3% (~15.55%) (Weiss et al. [10]) Pre-pubertal SGA: 4.8% AGA: 4.8% Pubertal SGA: 10.8% AGA: 7.3% Adolescents SGA: 26.3% AGA: 16.8%	n/d	n/d	n/d	n/d	n/d	n/d
Pelin, A.M., 2012 [22]	Romania	120 (61/59) Obese	7–18	n/d	120 (100%), of whom 35 (29.2%) with severe obesity	67 (55.8%) Obese (IDF)	-	67 (100%) F: 29 M: 54	67 (100%)	32 (47.76%)	33 (49.25%)	12 (17.9%)
González, M., 2012 [27]	Spain	362 (219/143)	12–17	16.6% F: 15.1% M: 18.9%	6.1% F: 5.0% M: 7.7%	21 (5.8%) (De Ferranti 2004 [12])	F: (2.7%) M: (10.5%)	n/d	n/d	n/d	n/d	n/d
Civantos Modino, S., 2012 [29]	Spain	148 (70/78) Two groups: - with uric acid ≥ 5.5 mg/dL - with uric acid < 5.5 mg/dL	5–19	4.8%	94%, of whom 96.9% with severe obesity	19.6% (IDF) Group with uric acid ≥ 5.5 mg/dL: 57% Group with uric acid < 5.5 mg/dL: 43%	n/d	n/d	n/d	n/d	n/d	n/d

Abdominal (Ab), waist circumference (WC), blood pressure (BP), systolic blood pressure (SBP), diastolic blood pressure (DBP), females (F), high-density lipoprotein (HDL), males (M), no data (n/d), number of participants (*n*), metabolic syndrome (MetS), triglycerides (TG), small for gestational age (SGA), appropriate for gestational age (AGA).

**Table 3 nutrients-15-03895-t003:** Critical appraisal of studies reporting the epidemiological burden of MetS in overweight and obese European children and adolescents [15].

References of the Studies Evaluated	Q1	Q2	Q3	Q4	Q5	Q6	Q7	Q8	Total (%) Yes
Smetanina, N., 2021 [16]	yes	yes	yes	yes	Not applicable	Not applicable	yes	yes	75
Jurkovičová, J., 2021 [23]	yes	yes	yes	yes	Not applicable	Not applicable	yes	yes	75
Jankowska, A., 2021 [17]	yes	yes	yes	yes	Not applicable	Not applicable	yes	yes	75
Leone, A., 2020 [24]	yes	yes	yes	yes	Not applicable	Not applicable	yes	yes	75
Wang, J., 2019 [26]	yes	yes	yes	yes	Not applicable	Not applicable	yes	yes	75
Szabelska-Zakrzewska, K., 2018 [18]	yes	yes	yes	yes	Not applicable	Not applicable	yes	unclear	62.5
Ostrihoňová, T., 2017 [25]	yes	yes	yes	yes	Not applicable	Not applicable	yes	no	62.5
Galera-Martínez, R., 2015 [19]	yes	yes	yes	yes	Not applicable	Not applicable	yes	yes	75
González-Jiménez, E., 2015 [28]	yes	yes	yes	yes	Not applicable	Not applicable	yes	yes	75
Martino, F., 2015 [13]	yes	yes	yes	yes	Not applicable	Not applicable	yes	yes	75
Pastucha, D., 2014 [20]	yes	yes	yes	yes	Not applicable	Not applicable	yes	unclear	62.5
Stroescu, R., 2014 [21]	yes	yes	yes	yes	Not applicable	Not applicable	yes	unclear	62.5
Pelin, A.M., 2012 [22]	yes	yes	yes	yes	Not applicable	Not applicable	yes	unclear	62.5
González, M., 2012 [27]	yes	yes	yes	yes	Not applicable	Not applicable	yes	unclear	62.5
Civantos Modino, S., 2012 [29]	yes	yes	yes	yes	Not applicable	Not applicable	yes	unclear	62.5
Total (%) Yes	100	100	100	100	-	-	100	53.3	--

Q1: Were the criteria for inclusion in the sample clearly defined? Q2: Were the study subjects and the setting described in detail? Q3: Was the exposure measured in a valid and reliable way? Q4: Were objective, standard criteria used for measurement of the condition? Q5: Were confounding factors identified? Q6: Were strategies to deal with confounding factors stated? Q7: Were the outcomes measured in a valid and reliable way? Q8: Was appropriate statistical analysis used?

## Data Availability

Not applicable.

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
