# Peer review of "Epidemiological Impact of Metabolic Syndrome in Overweight and Obese European Children and Adolescents: A Systematic Literature Review"

_nutrients, 2023, doi:10.3390/nu15183895_

Round 1
Reviewer 1 Report
I was honored to review the manuscript. The study presents high quality and deals with important clinical issues, such type of study is needed. I have only a few small remarks that the authors should address properly.
I recommend accepting the manuscript after minor revision.
There are only some points to correct:
- the aim is not clearly specified, although it is understandable when reading the whole article. Could You add one clear sentence about the intention, a problem that the article is trying to solve? Maybe a hypothesis, which will be confirmed or not in the conclusion section?
- please provide the list of abbreviations
- - introduction and discussion section need improvement; please provide information on how your results will translate into clinical practice;
- in the discussion section please provide the study's strong points and study limitation section
- please correct typos
All the abovementioned issues are crucial for the credibility of the results. The paper can be accepted only after addressing all the issues and another subsequent review.
Author Response
Dear Editor and Dear Reviewer,
We would like to thank you for the opportunity to resubmit our work. We have revised the paper in accordance with the suggestions received and we hope that these changes have contributed to its improvement.
Best regards,
the Authors

Reviewer 2 Report
nutrients-2580615-peer-review-v1
Generally, I am not in favor of systematic reviews where authors scan available literature and digest and provide summary of the observed fact, however, present study sound interesting, since comparing the observation in different European countries and try to standardize criteria and to serve as guideline when obesity in children's and adolescents is discussed. However, maybe some additional paragraphs in this directions can be provided by the authors.
L15-30: Maybe authors can consider correcting the abstract and do it as one flowing text, and do not having subsections. Maybe some additional, more specific data can be added to the abstract.
L77: please, correct to "population"
L95: Please, word in Lattin need to be in italics. Example "In vitro", etc.
Maybe if authors can introduce kind of recommendations for the standardizing the criteria and values for the obesity definition, including values for the biological markers will be a good addendum to the paper. In the paper was clearly mentioned that such criteria are generally adopted from the adults, but specific definition for children’s and adolescents need to be introduced.
Maybe some historical visual material (histograms) can be introduced to show how the obesity was changed in the years, and between different European countries, and maybe a short paragraph comparing with other continents can be introduced.
Additional worry is regarding the ethnic belonging of analyzed groups of children. All of them where of Caucasian origin? Study can be conducted in Spain (for examples) but can be including high number of the children’s from not Caucasian origin. Any information about this point?
Author Response
Dear Editor and Dear Reviewer,
We would like to thank you for the opportunity to resubmit our work. We have carefully incorporated the recommendations provided, aiming to enhance the quality of our work.
Best regards,
The Authors
